# The Influencing Mechanism of Incidental Emotions on Risk Perception: Evidence from Event-Related Potential

**DOI:** 10.3390/brainsci13030486

**Published:** 2023-03-13

**Authors:** Shu Zhang, Xinrou Yu, Xiuzhi Shi, Yan Zhang

**Affiliations:** 1School of Resources and Safety Engineering, Central South University, Changsha 410083, China; 2School of Educational Science, Huazhong University of Science and Technology, Wuhan 430074, China

**Keywords:** risk perception, incidental emotions, event-related potentials (ERPs), information-processing model, P2, N2, LPP

## Abstract

Incidental emotions would lead to accidents by influencing risk perception. However, few works of research further studied how incidental emotions affect risk perception at the neurological level. Before the experimental task, we used video clips for emotion elicitation. Then, the event-related potential (ERP) technique was used to obtain data on the risk perception process. The results showed that: compared to neutral emotion, the participants’ average reaction time was significantly shorter in positive and negative incidental emotion. Under negative incidental emotion, individuals overestimated risk and had a more significant deviation in risk perception; the amplitude of P2 and N2 components increased, and the amplitude of LPP component decreased under negative incidental emotion. Under positive incidental emotion, individuals’ correct-response rate was higher. These findings indicated that incidental emotions affected the mid-term risk analysis stage and the late risk judgment stage of risk perception. In the mid-term risk analysis stage, individuals processed high-risk information with a negativity bias which led to stronger cognitive conflict, while individuals assessed risks more accurately due to a larger attentional span under positive incidental emotions. In the late risk judgment stage, individuals under negative incidental emotion devoted few attentional resources to risk information which led to a risk judgment deviation. In contrast, individuals had a more detailed cognitive process of risk information under positive incidental emotion. On these bases, this paper confirmed the influence of incidental emotions on risk perception and established an emotional information-processing model. This study provided a reference for emotional interventions to facilitate accident prevention.

## 1. Introduction

Studies have demonstrated that emotions are one of the important factors contributing to accidents [1,2,3]. For example, in 2011, a nuclear leak in Japan sparked public panic and the subsequent ‘salt rush’ [4], while in 2021, in a hit-and-run car accident in Dalian, China [5], the driver lost control of his emotions owing to investment failure and generated retaliatory social behavior, resulting in five deaths and five injuries. Accidents caused by incidental emotions resulted in bad social impacts and serious damage to people and property. Individuals with different emotions make varying behavioral decisions, such as individuals under angry emotions choosing a risk-taking approach and displaying risk-seeking behavior [6]. In the risk-as-feelings model [7], emotions can also indirectly influence behavioral decisions by influencing cognitive assessment processes. For example, in research on driving behavior decisions, scholars have found that emotions affected the driver’s attention [8], subjective feelings [9], and the ability to recognize hazards [10], and then affected the outcome of behavioral decisions. Thus, it is necessary to study the role of incidental emotions on risk perception. Emotion was one of the main factors influencing risk perception [11,12,13,14]. Thus, it can be seen that studying the influence of incidental emotions on risk perception and exploring the brain mechanisms in which incidental emotions affect risk perception are of great significance for preventing emotionally induced accidents.

Incidental emotions are the emotions experienced by individuals that originated from objective situations unrelated to the current task [15]. Although unrelated to the task, incidental emotions significantly affect cognitive and decision-making behaviors [16,17]. Positive emotions reduced the perceived probability of risky events, while negative emotions did the opposite [18,19]. Positive emotions also triggered automatic processing that relied on prototypes and schemas, and individuals paid more attention to details under negative emotions [20]. Several studies have found that individuals under positive emotions showed flexibility in thinking, and that positive emotions facilitated cognitive processes [21,22]. Under negative emotions, individuals engaged in more thoughtful cognitive processing [23,24]. Negative emotions also interfered with behavioral inhibition processes and diminished individuals’ behavioral control, as evidenced by enhanced brain activity for inhibitory processing and prolonged processing time [25]. Another fMRI study has shown that disturbances with negative emotions lead to poorer cognitive performance [26]. Current research suggests positive emotions may promote cognitive processing, while negative emotions may interfere with cognitive processing [27]. Positive emotions reduced the perceived probability of risky events, while negative emotions did the opposite. These findings provided a certain basis for our research hypothesis. However, few studies explained the process of the influence of incidental emotions on risk perception. Therefore, based on previous research, we explored different processes of risk perception under different incidental emotions (positive/negative).

Scales, interviews, or behavioral experiments [28,29] were mostly used in current studies related to risk perception. However, those methods are more reflective of behavioral tendencies but lack the reflection of internal mental processes. With the advantages of high temporal resolution and non-invasiveness, the event-related potential (ERP) technique can provide an objective physiological and psychological basis for cognitive studies and has been applied by some scholars in risk perception studies [30,31,32,33]. Based on the communication and human information processing (C-HIP) model proposed by Wogalter and colleagues [34,35], Ma et al. [30,33] proposed the hazard perception two-stage model (HPTS) by an ERP experiment. They characterized cognitive processes including attention and evaluation, through ERP component indicators such as P2 and LPP. In ERP studies, components including N2, P2, and LPP were mainly analyzed and discussed for risk perception. Negative stimuli evoked a larger P2 amplitude compared to positive and neutral stimuli, indicating the attentional bias of negative stimuli [36]. In contrast, a study found that negative stimuli may evoke a lower P2 amplitude than positive and neutral stimuli [37]. These opposing findings suggest that P2 reflects an attentional bias: the more attention resources allocated, the larger the amplitude of P2 [38]. Correll et al. [39] found that pictures of unarmed figures elicited a larger N2 amplitude than that of armed figures, reflecting a response suppression process for unarmed figures. Studies found that stimuli with threatening properties induce a larger LPP amplitude than neutral or positive stimuli [30,33,40,41]. Risky events elicit a larger LPP amplitude than safe events [42,43,44]. In addition, compared to neutral pictures, positive and negative pictures evoke a significantly enhanced LPP amplitude [45]. Moreover, the processing of emotional stimuli is a mandatory process, manifested as an enhanced LPP amplitude [46,47]. Although there were many ERP studies related to risk perception, few studies focused on the influence of incidental emotions on risk perception. However, previous studies still provided the basis for our research hypothesis. In addition, different risk events (high/low) may affect the risk perception processes, mainly reflected in changes in the LPP components. In this study, the effects of high and low risk and incidental emotions on risk perception might be co-existent.

In summary, incidental emotions would lead to accidents by influencing risk perception. This study employed ERP experiments further to explore the influence of incidental emotions on risk perception. Positive and negative incidental emotions were used as the independent variables. Additionally, an emotional information-processing model was created to help understand the role of incidental emotions in risk perception. Ultimately, this study aimed to serve as a reference to support the ratio assessment of risk and provide guidance for preventing accidents rooted in emotions.

As mentioned above, we hypothesized that positive incidental emotions might promote the underestimation of risk, manifested as higher P2 and LPP amplitudes. Negative incidental emotions may lead to the overestimation of risk, manifested as an increased P2 amplitude and LPP amplitude relative to a neutral sentiment and an increased N2 amplitude in relatively positive incidental emotion.

## 2. Materials and Methods

### 2.1. Participants

Twenty postgraduate students (seven women), ranging in age from 22 and 24 years (23.300 ± 0.571, Mean ± SD), were recruited from Central South University. They did not participate in the pre-test of emotional and picture stimuli evaluation. All participants were right-handed, had normal or corrected-to-normal visual acuity, and did not have any history of neurological or mental diseases. They provided informed consent and were paid for participation.

### 2.2. Materials

#### 2.2.1. Emotion-Inducing Stimuli

Videos were used to induce incidental emotion [48]. Twelve film clips lasting 2–4 min separately induced different valence of emotions (positive/negative). The self-assessment manikin (SAM) [49] measured the valence and arousal of emotions induced by film clips. Twenty-five people who did not participate in the electrophysiological experiment rated valence and arousal degrees of emotion induced by film clips. The stimuli were pictures depicting scenes that contained common risks from public society. After watching the film clips, they were asked to fill out the SAM scale, which consisted of two columns of questions: the first column described emotions, ranging from positive (a smiling figure) to neutral, to negative (a frowning figure), rated on a scale of 9–1 from positive to negative; the second column rated arousal intensity, ranging from high-intensity arousal (eyes wide open, representing the explosive blast sign) to low-intensity arousal (eyes closed, representing the small pinprick sign), again rated on a scale of 9–1 from high-intensity arousal to low-intensity arousal.

Five film clips with better induction effects were selected for the different valence of inducing emotions. The selected positive emotion films were ‘Goodbye Mr. Loser’, ‘The Mermaid’, ‘Detective Chinatown 2′, and ‘Never Say Die’, and the negative emotion film was ‘Silenced’. Paired *t*-test showed a significant difference in mean valence score between the two groups [M _negative_ = 3.506, M _positive_ = 8.408, t (4) = 23.90, *p* < 0.001, Cohen’s *d* = 23.9].

#### 2.2.2. Picture Stimuli

The stimuli were pictures depicting scenes that contained common risks from public society. The 172 people who did not participate in the electrophysiological experiment assessed the risk degree of pictures, according to the LEC method [50]. For each picture, they were asked two questions: ‘What do you think is the probability of an accident in the scenario presented in this picture?’ and ‘What do you think is the degree of injury that would result from the accident?’. There were five options for each of the two questions. The ‘probability of an accident’ rating scores from low to high are: very unlikely (0.5), unlikely (1), likely (3), quite likely (6), and completely likely (10). The ‘degree of injury caused by the accident’ rating scores from low to high are: attention-grabbing (3), minor injury (7), serious (15), quite serious (40), and extremely serious (100). The risk degree of each picture was the product of the mean scores of the two questions. Fifty pictures were selected and classified as the high and low groups by risk degree (30% high group, 70% low group). The *t*-test results showed a significant difference between the two groups of the mean risk levels [M _high group_ = 336.469, M _low group_ = 137.111, t (17) = 1.740, *p* < 0.001, Cohen’s *d* = 0.844].

### 2.3. Procedure

Participants were seated in a quiet lab to complete the study, including three oddball paradigm experiments under three emotion states. Each experiment consisted of 5 blocks of 50 trials, with each block including 35 standard (low-risk) and 15 deviant (high-risk) pictures. Experiments were grouped as neutral emotion (NE), positive incidental emotion (PIE), and negative incidental emotion (NIE). Experiments were performed in the above order. The experiment under the neutral emotion group was conducted without watching any video and in the calm state of the participants. A film clip was played before each block in the PIE and NIE groups. Stimuli were presented on the computer screen at a distance of 100 cm with a visual angle of 6.817° × 5.759° (11.906 × 10.054 cm, width × height). As shown in Figure 1, each trial started with a black fixation cue (+) centered on a white background for 500 ms, followed by an empty screen for 500 ms. Then, a picture was presented at the center of the screen for 2500 ms against a white background, then an empty screen for 500 ms appeared. The order of pictures was random within a block. The break time between each block was five minutes. E-Prime 3.0 (Psychology Software Tools Inc., Zeppelinstraße, Gilching, GER) was used to write the experimental program.

The participants were told to press the SPACE key if they thought the risk degree of scenes presented by the picture was high, and they had no response to the low-risk pictures they indicated. Before and after the experiments, all participants were asked to conduct an interview survey to ensure the validity of emotion-inducing materials.

### 2.4. ERP Recording and Data Analysis

EEG data were recorded by an electrode cap (Brain Products, GER) with 32 Ag/AgCl electrodes mounted according to the extended international 10–20 system. Impedances were maintained below 5 kΩ throughout the experiment. ERP averages were computed offline by Analyzer 2.2.0 (Psychology Software Tools Inc., Zeppelinstraße, Gilching, GER). The arithmetic mean of bilateral mastoids (TP9/TP10) was first elected in Analyzer software to re-reference the raw EEG data, and the filter bandpass was set to 0.3–40 Hz. Trials with amplifier clipping artifacts or peak-to-peak deflection exceeding ±100 V were excluded from averaging. Artifacts due to cardiac activity, muscle movement, eye movement, and blinks were removed by independent component analysis (ICA). The ERP waveforms for each emotional state were obtained by segmentation according to the incidental emotions (neutral/positive/negative). Stimulus-locked data were segmented into epochs comprising 200 ms before stimulus onset and 800 ms after the onset. The baseline correction was carried out using the first 200 ms of each channel. After presenting high- and low-risk pictures, the ERP waveforms were screened for superimposed averaging, and the segmented ERP waveforms were grouped and averaged separately for each emotional state.

Three-factor repeated-measure analysis of variance (ANOVA) was conducted for the amplitude of ERP components. ANOVA factors were valence condition (three levels: NE, PIE, and NIE), risk degree (two levels: high-risk and low-risk), and electrode sites. Behavioral data were analyzed by one-way repeated-measure ANOVA on the number of keystrokes, correct-response rate, and the mean reaction time. The correct-response rate is the value of the correct response number divided by the number of keystrokes, and the mean reaction time is the value of the total reaction time divided by the number of keystrokes. The results of post hoc tests were corrected according to the Bonferroni method.

## 3. Results

### 3.1. Behavioral Data

For the number of keystrokes, a one-factor repeated-measure ANOVA showed a significant main effect of valance [F(2, 18) = 7.439, *p* = 0.004 < 0.01, η^2^ = 0.453]. Post hoc tests showed that the number of keystrokes was significantly larger [*p* = 0.011 < 0.05, Cohen’s *d* = 0.210] in the PIE condition than in the NE condition. In addition, the number of keystrokes in the NIE condition was significantly larger than in the neutral and positive conditions [*p* = 0.003 < 0.01, Cohen’s *d* = 0.368; *p* = 0.018 < 0.05, Cohen’s *d* = 0.172]. For the correct-response rate, a one-factor repeated-measure ANOVA showed a significant main effect of valance [F(2, 18) = 5.479, *p* = 0.008 < 0.01, η^2^ = 0.224]. Post hoc tests showed that the correct-response rate was significantly larger [*p* = 0.019 < 0.05, Cohen’s *d* = 0.348] in the PIE condition than in the NIE condition. For the mean reaction time, a one-factor repeated-measure ANOVA showed a significant main effect of valance [F(2, 18) = 9.942, *p* < 0.001, η^2^ = 0.869]. Post hoc tests showed that the mean reaction time was significantly shorter [*p* < 0.001, Cohen’s *d* = 0.560] in the PIE condition than in the NIE condition. In addition, the mean reaction time in the NIE condition was shorter than in NE and PIE conditions [*p* < 0.001, Cohen’s *d* = 1.769; *p* < 0.001, Cohen’s *d* = 0.560]. All the behavioral data results are shown in Table 1.

### 3.2. ERP Data

The waveform maps and EEG topography after averaging, superimposed at different emotional states with different risk levels, were shown in Figure 2. Figure 2A shows the EEG waveform maps of the anterior and posterior parts of the scalp in different emotional states for different risk level pictures, respectively, using the Fz and P4 electrode sites as examples.

As shown in Figure 2A, the P150/N170, P2, N2, and LPP components were elicited in all experiments. Based on Figure 2A,B, it can be observed that the P150/N170 complex wave components with opposite polarity were evoked in the parietal and occipital, and frontal and central sites, respectively, within the latency period of 130–200 ms. The P2 component was evoked in the parietal and occipital sites during the latency period of 180–260 ms and was more pronounced in the parietal and occipital sites. Significant N2 components were evoked in the prefrontal to central sites during the latency period of 180–250 ms in different emotional conditions. LPP components were evoked in the central frontal, parietal, and occipital sites within 350–550 ms of latency. Thus, we selected electrodes (P3, P4, Pz, O1, O2, and Oz) in the parietal and occipital area for N170, electrodes (F3, F4, Fz, FC1, FC2, C3, C4, and Cz) in the frontal to the central area for P150, electrodes (P3, P4, Pz, O1, O2, and Oz) in the parietal and occipital area for P2, electrodes (F3, F4, Fz, FC1, FC2, C3, C4, and Cz) in the frontal–central area for N2, and electrodes (F3, F4, Fz, C3, C4, Cz, P3, P4, Pz, O1, O2, and Oz) in the frontal, central, parietal, and occipital area for LPP.

For all experiments, the amplitude (baseline to peak) of the P150/N170 (130–200 ms), P2 (180–260 ms), N2 (180–250 ms), and LPP (350–550 ms) components were measured and analyzed.

(1)N170 and P150 components

For the N170 time window, the 3 × 2 × 6 repeated-measure analysis of variance showed a significant main effect of risk degree [F(1, 19) = 6.379, *p* = 0.021 < 0.05, η^2^ = 0.251] and a significant main effect of the electrode [F(5, 15) = 8.532, *p* = 0.001 < 0.05, η^2^ = 0.740]. There was no significant main effect of valance [F(2, 18) = 1.007, *p* = 0.375 > 0.05, η^2^ = 0.050]. The N170 amplitude was larger [*p* = 0.021 < 0.05, Cohen’s *d* = 0.707] for high-risk pictures (−1.706 ± 0.750 μV) compared with low-risk pictures (−1.173 ± 0.757 μV).

For the P150 time window, the 3 × 2 × 8 repeated-measure analysis of variance showed a significant main effect of risk degree [F(1, 19) = 6.379, *p* = 0.021 < 0.05, η^2^ = 0.251]. There was no significant main effect of valance [F(2, 18) = 0.640, *p* = 0.539 > 0.05, η^2^ = 0.066] and electrode [F(5, 15) = 30.562, *p* < 0.001, η^2^ = 0.617]. The P150 amplitude was larger [*p* < 0.001, Cohen’s *d* = 0.541] for low-risk pictures (0.721 ± 0.491 μV) compared with high-risk pictures (−0.472 ± 0.428 μV).

(2)P2 component

For the P2 time window, the 3 × 2 × 6 repeated-measure analysis of variance showed a significant main effect of valence [F(2, 18) = 4.433, p = 0.027 < 0.05, η^2^ = 0.330] and a significant main effect of the electrode [F(5, 15) = 9.454, *p* < 0.001, η^2^ = 0.759]. There was no significant main effect of risk degree [F(1, 19) = 0.336, *p* = 0.569 > 0.05, η^2^ = 0.017]. The amplitude of P2 induced in the NIE condition (7.356 ± 0.886 μV) was larger [*p* = 0.024 < 0.05, Cohen’s *d* = 0.926] than that in the PIE condition (6.558 ± 0.837 μV). The interaction of risk degree × electrode on P2 was significant [F(5, 15) = 3.142, *p* = 0.039 < 0.05, η^2^ = 0.512]. A simple effects analysis showed that the P2 amplitude was larger [*p* = 0.032 < 0.05, Cohen’s *d* = 0.651] induced by high risk (6.981 ± 0.815 μV) than low risk (6.454 ± 0.803 μV) at the P4 electrode site. The interaction of valance × risk degree × electrode on P2 was significant [F(10, 10) = 6.902, *p* = 0.003 < 0.05, η^2^ = 0.873]. A simple effects analysis showed that the P2 amplitude evoked by high-risk pictures in different conditions differed significantly at the Oz and O2 electrode sites. In addition, the P2 amplitude evoked by pictures in the NIE condition was significantly larger than in the PIE condition [*p* = 0.020 < 0.05, Cohen’s *d* = 1.158; *p* = 0.024 < 0.05, Cohen’s *d* = 1.286]. For low-risk pictures, the P2 amplitude in different conditions differed significantly at the O1, Oz, and O2 electrode sites. Moreover, the P2 amplitude in the NIE condition (9.097 ± 0.935 μV) was larger than that in the PIE condition (8.421 ± 0.909μV) at the O1 electrode site [*p* = 0.014 < 0.05, Cohen’s *d* = 0.733]. The P2 amplitude in the NIE condition was significantly larger than that in NE [*p* = 0.043 < 0.05, Cohen’s *d* = 1.400; *p* = 0.024 < 0.05, Cohen’s *d* = 1.641] and PIE [*p* = 0.049 < 0.05, Cohen’s *d* = 1.061; *p* = 0.043 < 0.05, Cohen’s *d* = 1.253] condition at the Oz and O2 electrode sites. At the P4 electrode site, the P2 amplitude was significantly larger [*p* = 0.007 < 0.01, Cohen’s *d* = 1.106] induced by high risk (7.752 ± 0.737 μV) than low risk (6.862 ± 0.867 μV) in NIE condition, as shown in Table 2.

(3)N2 component

For the N2 time window, the 3 × 2 × 8 repeated-measure analysis of variance showed a significant main effect of valence [F(2, 18) = 3.673, *p* = 0.046 < 0.05, η^2^ = 0.290] and a significant main effect of the electrode [F(7, 13) = 9.454, *p* = 0.001 < 0.05, η^2^ = 0.816]. There was no significant main effect of risk degree [F(1, 19) = 1.824, *p* = 0.193 > 0.05, η^2^ = 0.088]. None of the interaction effects were significant. The amplitude of N2 induced by pictures in the NIE condition (−5.525 ± 0.432 μV) was significantly larger [*p* = 0.042 < 0.05, Cohen’s *d* = 1.640] than that in the PIE condition (−4.806 ± 0.445 μV).

(4)LPP component

For the LPP time window, the 3 × 2 × 12 repeated-measure analysis of variance showed a significant main effect of valence [F(2, 18) = 3.686, *p* = 0.046 < 0.05, η^2^ = 0.291] and a significant main effect of the electrode [F(11, 9) = 9.454, *p* = 0.017 < 0.05, η^2^ = 0.844]. There was no significant main effect of risk degree [F(1, 19) = 0.025, *p* = 0.875 > 0.05, η^2^ = 0.001]. None of the interaction effects were significant. The mean amplitude of LPP was significantly smaller [*p* = 0.039 < 0.05, Cohen’s *d* = 3.585] in the NIE condition (−11.355 ± 4.264 μV) than that in the NE condition (0.438 ± 0.607 μV). In addition, the mean amplitude of LPP in neutral emotion was not significantly different from that in the PIE condition (8.476 ± 14.142 μV).

All the ERP components’ results are shown in Table 3.

## 4. Discussion

### 4.1. Behavioral Effects

The results revealed that both positive and negative incidental emotions increased brain arousal. The mean reaction time was significantly shorter for both the PIE and NIE conditions compared to the NE condition, with the NIE condition having the shortest mean reaction time. This result was in line with other research findings [45]. Changes in the perceived speed of high-risk pictures may be related to the processing dominance in brain regions. Brain-imaging studies have shown that both positive and negative emotions have specific brain-processing systems, with a right hemisphere dominance when processing negative emotions, and a left hemisphere dominance when processing positive emotions [51]. As a result, different levels of emotional responses triggered by pictures of different risk levels were processed more in the right brain. We suggested that there was a processing bias towards high-risk pictures under negative incidental emotion relative to positive incidental emotion.

Incidental emotions can also influence risk perception activities. Compared with NE conditions, the number of keystrokes increased significantly in both the PIE and NIE conditions, with the most keystrokes in the NIE condition. The results showed that positive and negative incidental emotions led to varying degrees of risk perception bias. Previous studies have found similar results. Hou [52] found that using the task-shifting paradigm under positive and negative emotional conditions had a switching cost, suggesting that both positive and negative incidental emotions interfere with cognitive activity. We suggested that incidental emotions, as thinking activities unrelated to the task at hand, occupy limited cognitive resources and thus affect individuals’ performance in the task [53,54]. It is also indicated that negative incidental emotion leads to overestimating risk, which supports the affect heuristic theory [19]. Previous studies have found similar results. Sobkow et al. [55] found that participants under stress rated the perceived risk higher than those under control. Joen et al. [56] found that participants perceived higher risk under induced anger, and estimated that the reason was that anger encouraged a stereotypic thought process. Fredrickson et al. [57] suggested that negative incidental emotions interfered with cognitive activities by narrowing the scope of attention and response. As a result, we considered that negative incidental emotions narrowed the scope of attention, and led to a negativity bias toward high-risk pictures, which resulted in overestimating risk.

In the PIE condition, the number of keystrokes increased relative to the NE condition, while the correct-response rate was larger than in the NIE condition. Previous studies have found similar results. Dreisbach et al. [58] found that switching an individual’s response to a particular color under the PIE condition was less costly than under the NE condition. Then, they suggested that positive emotion reduced the impact on innate response patterns and increased sensitivity to new stimuli, thus promoting cognitive flexibility. We considered that this phenomenon might also be task-related. Studies have shown that positive emotions increase attention span, and individuals tend to use holistic and divergent behavior patterns, increasing cognitive flexibility [56]. However, positive incidental emotion interfered with the cognitive activities of detailed processing when the task required detailed processing [59]. Accordingly, we considered that the scope of attention and sensitivity to deviant stimuli increased under positive incidental emotion. Individuals were more likely to accurately judge high-risk pictures, while risky pictures requiring careful identification can go awry.

In short, incidental emotions can affect risk perception, even if they are not task-related. Positive and negative incidental emotions cause increased brain arousal, and there was a processing advantage for risk perception under negative incidental emotions. Incidental emotions occupied limited cognitive resources and thus affected individuals’ performance on the task. Negative incidental emotion interfered with risk perception processes and led to overestimating risk. Positive incidental emotions increased the scope of attention, and individuals were more likely to judge risk accurately.

### 4.2. Neural-Level Activities of Risk Perception

#### 4.2.1. Construction of Emotional Information-Processing Model

In the experiment, there were several cognitive activities. Participants needed to perceive whether it was risky or not according to information from pictures. Then, they needed to combine the text and picture information to analyze the risk scenarios, such as the types of hazards in the risky scenarios, the probability of accidents, and the severity of injuries caused by the accidents. Moreover, they also needed to judge and evaluate the level of risk. To objectively interpret the behavioral outcomes, this study used the P150/N170, P2, N2, and LPP components collected in the experiment as neural indicators of risk perception. Particularly, the information processing of risk perception in this study was divided into three stages that were the early hazard perception stage, indicated by P150/N170, the mid-term risk analysis stage, indicated by P2 and N2, and the late risk judgment stage, indicated by LPP. Based on previous theories and the results of this study, the emotional information-processing (EIP) model was proposed, as shown in Figure 3.

##### The Early Hazard Perception Stage

In the early hazard perception stage, hazard identification is the main process. During this stage, the P150/N170 components emerged. The P150/N170 complex wave is defined as a positive component of the anterior and middle electrodes and a negative component of the occipital electrodes [60]. The N170 component is associated with visual attention [61] and involves the low-level processing of early vision. Therefore, the P150/N170 complex wave can indicate the early information perception stage. The results showed that this stage was not affected by incidental emotions.

Incidental emotions, as thinking activities unrelated to the current task, would take up certain cognitive resources while increasing the individuals’ arousal, thus impacting the individuals’ subsequent task performance. Incidental emotions did not affect the early hazard perception stage in the risk perception process. Individuals quickly searched for high-risk information due to survival instincts, known as negativity bias.

##### The Mid-Term Risk Analysis Stage

In the mid-term risk analysis stage, individuals perform accident consequence and probability analyses. At this stage, the P2 component was more evident. The P2 component was often found in ERP studies of complex visual stimuli and may be involved in the early semantic processing of visual information [62]. The P2 component was evoked in the parietal and occipital areas in the present experiment. The results showed that incidental emotions significantly affected the P2 amplitude in the parietal and occipital areas. N2 reflects cognitive control processes that inhibit attention or strong responses [37], and the N2 amplitude implied the number of control resources invested in resolving cognitive conflicts [63]. The results showed that the N2 amplitude was influenced by incidental emotion.

In the mid-term risk analysis stage, the attention span to risky information of individuals under positive incidental emotion is larger than that in negative incidental emotion. In the same period, an attentional bias oriented by positive working memory and task demand partially conflicts with the low-risk stimuli in the experiment, producing a smaller risk analysis deviation. In addition, individuals invest some cognitive control resources by keystroking to resolve the cognitive conflict, resulting in a shorter reaction time and a larger number of keystrokes than under neutral emotion. Under negative emotion, individuals developed an attentional processing bias to high-risk information in the risk analysis stage. Then, the attentional bias oriented by negative working memory, high-risk attentional processing bias, and task demand created stronger cognitive conflict with most low-risk stimuli. Correspondingly, individuals invested more cognitive control resources by keystroking to quickly resolve conflicts, resulting in more keystrokes and a shorter reaction time.

##### The Late Risk Judgment Stage

LPP is a late positive component reflecting the active engagement of attentional resources and an endogenous evaluative processing to a stimulus task [41,64]. Therefore, the LPP component was used in this study to describe the late stage of risk judgment.

In the late risk judgment stage, individuals may process risk information more exhaustively under positive incidental emotion, and the risk judgment deviation was relatively less. Those might lead to a larger correct-response rate than negative incidental emotion, and individuals were more likely to assess risk accurately. Under negative incidental emotion, individuals did not pay enough active attention to risk information, and the cognitive deviation in the risk analysis stage was not corrected in time. This led to a larger risk judgment deviation, shorter reaction time, lower correct response rates, and a tendency to overestimate risk.

#### 4.2.2. Neural Activities of Incidental Emotions Affecting Risk Perception

The early hazard perception stage mainly included the automatic and rapid detection of risk information, with the performance components of P150 and N170, which were sensitive to both high and low risks. Compared with low-risk pictures, high-risk pictures evoked a larger N170 amplitude. This implied a negativity bias in individuals’ early perceptual processing of relatively high-risk information (deviant stimuli). In addition, the N170 amplitude varied significantly in the left parietal and occipital area, suggesting a higher visual perception of high-risk pictures. Geng et al. [65] found that self-related stimuli evoked a lower prefrontal P150 amplitude than other stimuli, suggesting the rapid, low-resource-consuming self-related processing. In this study, high-risk pictures evoked a lower P150 amplitude than low-risk pictures, implying the rapid and low-resource-consuming processing for risk information in high-risk pictures. Humans were more quickly aware of threat information after a long period of evolution due to survival instincts, which was known as negativity bias. In addition, individuals have a perceptual processing bias for high-risk information and can automatically and quickly identify high-risk stimuli.

The mid-term risk analysis stage mainly included subjective attention to risk events and conflict analysis, and the expression components were P2 and N2, which were sensitive to incidental emotions. For the same level of pictures, a larger P2 amplitude was evoked under the NIE condition than under other conditions. The result proves that negative incidental emotion interferes with the risk analysis stage. It indicates that negative incidental emotions consume more attentional resources that are irrelevant to the task, reducing task-related attentional resource processing, and resulting in a risk analysis deviation, such as the analysis of hazard type, likelihood, and degree of risk. In addition, the P2 amplitude in the NIE condition was larger than the P2 under the PIE condition, indicating that negative incidental emotion consumed more attention resources for the same task than positive incidental emotion [66]. Moreover, the results revealed that individuals under negative incidental emotion have a negativity bias to high-risk stimuli. The P2 amplitude evoked by high-risk pictures was larger than that evoked by low-risk pictures in the NIE condition. In contrast, there was no significant difference between P2 magnitudes evoked by high- and low-risk pictures in the PIE condition. It indicated that the visual attentional processing of high-risk information (negative stimuli) dominated in individuals under negative incidental emotion relative to neutral and positive incidental emotions. In other words, high-risk information was easier to obtain during attentional processing under negative incidental emotion. Furthermore, the results showed a more pronounced attentional processing advantage in the right parietal area, with similar results in Guo et al. [67]. This finding added to our belief that individuals have a processing advantage in dealing with risky stimuli under negative incidental emotion. Taken together, in the mid-term risk analysis stage, individuals under negative incidental emotion allocated less attentional resources to the task and had an attentional processing bias for high-risk information. This finding suggested that negative incidental emotion narrowed the span of attention and response range of individuals, which may result in a lower correct-response rate for individuals. In this study, positive incidental emotion consumed fewer cognitive resources and a greater attention span to risk information than negative incidental emotions. This might be the root cause of the larger correct response under positive incidental emotion. A larger N2 amplitude in the frontal area was induced by high-risk pictures under the NIE condition than under the PIE condition. This finding was consistent with Guo et al. [67], suggesting that negative incidental emotion led to stronger cognitive conflict. According to the visual attention theory [68], individuals were influenced by the risk information and the moderating effect of working memory [69] and task demands during risk analysis. Thus, risk information highly relevant to task demands and corresponding working memory templates was more likely to be noticed.

The late risk judgment stage mainly included risk assessment and judgment processes, and the performance component was LPP, which was sensitive to incidental emotions. LPP is involved in the evaluation classification process that can distinguish between different categories of stimuli in the active evaluation task [70]. The results showed that the mean LPP amplitude evoked by risk pictures was significantly lower under the NIE condition than under the NE condition, consistent with the findings of Cheng et al. [71]. It suggests that individuals devote fewer active attentional resources to risk information under negative incidental emotions than other conditions. It means that less judgmental processing of risk information is assessed, which generates a risk judgment deviation and leaves the deviation of risk analysis uncorrected. As a result, individuals under negative incidental emotion tended to overestimate risk and had shorter reaction time than under other conditions, which further confirmed the bias of individuals under negative incidental emotion to assess risk quickly and automatically. In contrast, the mean LPP amplitude did not differ significantly between the positive condition, and the other conditions were not significant. However, the larger mean LPP amplitude under the PIE condition may indicate a relatively more detailed cognitive process of risk information in individuals under positive incidental emotion. This may be an important reason for the larger correct-response rate under positive incidental emotion.

### 4.3. Implications of Findings

Theoretically, the present study provided a new perspective on the role of incidental emotions in the risk perception process. After that, the study revealed incidental emotions’ behavioral and neurological effects on risk perception. The emotional information-processing model was proposed, which included three stages of the risk perception process: the early information perception stage, the mid-term risk analysis stage, and the late risk judgment stage. Meanwhile, incidental emotions affected both the mid-term and late stages of risk perception. Finally, this study aimed to understand how incidental emotions affect the brain’s response to risk.

Emotion management is the focus of public safety management. Managers and regulators of individual and social emotions can enhance good mental health and improve social governance. The simulated daily emotions and realistic dangerous scenarios in this experiment also gave more practical significance to the results. First, maintaining a smooth emotional state is the cognitive basis for performing safe behaviors, and individuals should pay attention to the application of emotion management in their daily life. As a bad emotion, negative incidental emotion not only seriously impairs individuals’ mental health but also may lead to irrational behavior, resulting in serious accidents. Therefore, the timely channeling and resolving of negative incidental emotions are the focus of emotion management in public safety management. For the safety of enterprises, companies should identify and intervene in employees’ negative emotions to reduce employees’ risk-taking behavior. Appropriate positive incidental emotions can promote an accurate assessment of environmental risks by individuals. Therefore, maintaining positive incidental emotion facilitates a rational assessment of risks and thus prevents accidents from occurring.

### 4.4. Limitation and Future Research

There are some limitations that should be acknowledged. Coaster et al. [32] indicated that risk assessments for physical harm engaged a broad network of brain regions, which were sensitive to the severity of harm, the likelihood of risk, and the framing of risk information. This study examined the cognitive processes for high and low risk, but risk perception of different risk types or more refined risk classifications, such as high, medium, and low risk, were recommended for further investigation. Individual characteristics, such as gender, age, education level, income, and life experience (disaster experience) [11], somewhat influence risk perception. The participants in the present experiment were between 20 and 30 years old and had college degrees. Although the effects of other variables were controlled as much as possible by selecting college students as participants in the present experiments, a more diverse sample should be included to increase the validity of the results.

## 5. Conclusions

Different incidental emotions have varying effects on risk perception. Appling the ERP technique, we sought to explore the influence of incidental emotions on risk perception at a neurological level. We found that individuals with negative incidental emotion detected risk quickly and automatically, and processed risk information with a negativity bias, manifesting overestimated risk. Individuals with positive incidental emotions assessed risk more precisely. Based on these findings, we proposed an emotional information-processing model including the early information perception stage, mid-term risk analysis stage, and late risk judgment stage. We suggested that incidental emotions influenced two stages of the risk perception process: the mid-term risk analysis stage and the late risk judgment stage. This study provided objective evidence for behavioral testing studies to better understand how incidental emotions influence risk perception, and offered a reference for the emotional interventions to facilitate accident prevention.

## Figures and Tables

**Figure 1 brainsci-13-00486-f001:**
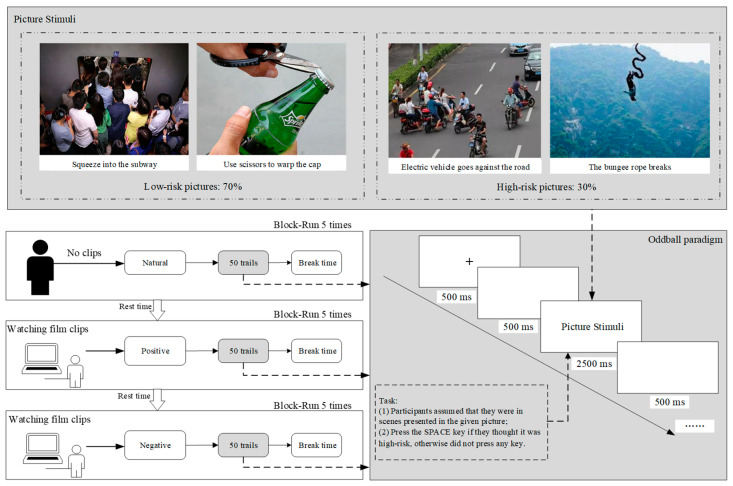
Experimental procedure.

**Figure 2 brainsci-13-00486-f002:**
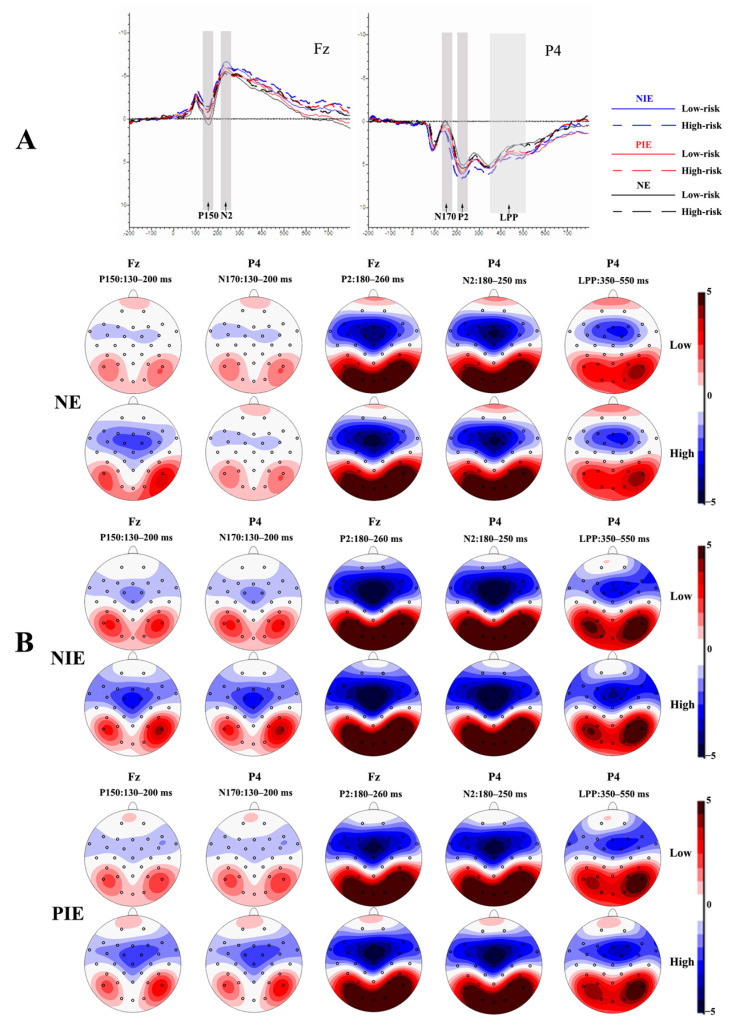
Event-related potential (ERP) results. (**A**) The averaged waveforms in frontal and parietal regions under neutral, positive, and negative conditions (with Fz and P4 electrode sites as examples); (**B**) the topographic maps of P150/N170, P2, N2, and LPP for different conditions. P2 is the second positive component of ERPs; N2 is the second negative component of ERPs; abd LPP is a negative component of ERPs observed at around 400 ms. Fz is a frontal midline scalp electrode, and P4 is a right-parietal region scalp electrode.

**Figure 3 brainsci-13-00486-f003:**
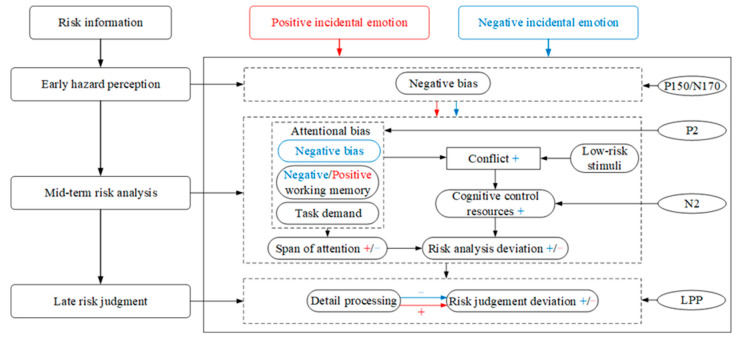
The emotional information-processing model. The “+” symbol represents enhanced effect, and the “−” symbol represents weakened effect. The blue font represents negative incidental emotion, and red font represents positive incidental emotion.

**Table 1 brainsci-13-00486-t001:** The behavioral data results.

Behavioral Variables	Emotional States (M ± SD μV)			F Test	
	NE	PIE	NIE	F Value	*p*
Number of keystrokes	152.550 ± 44.834	161.950 ± 44.749	169.650 ± 48.066	7.439	0.004 **
Correct-response rate	0.379 ± 0.051	0.382 ± 0.058	0.362 ± 0.057	5.479	0.008 **
The mean reaction time	1079.544 ± 153.853	873.274 ± 198.719	760.889 ± 202.953	9.942	0.000 ***

Notes: ** *p* < 0.01, *** *p* < 0.001.

**Table 2 brainsci-13-00486-t002:** The repeated-measure analysis of variance results of P2 amplitude.

Conditions	Risk Level (High/Low)	Electrodes (Mean ± SD μV)				Pairwise Comparison Results
		O1	Oz	O2	P4	
NE	High	8.380 ± 0.959	8.148 ± 1.011	9.278 ± 1.022	6.450 ± 0.978	(a) High-risk pictures: Oz, O2: NIE > PIE (*p* = 0.020 *, *p* = 0.024 *) (a) Low-risk pictures: O1: NIE > PIE (*p* = 0.014 *) (a) Low-risk pictures: Oz, O2: NIE > NE (*p* = 0.043 *, *p* = 0.024 *) NIE > PIE (*p* = 0.049 *, *p* = 0.043 *) (b) NIE condition: P4: High-risk > Low-risk (*p* = 0.007 **)
	Low	8.079 ± 1.000	7.717 ± 1.017	8.484 ± 1.014	5.845 ± 0.875	
PIE	High	8.421 ± 0.909	7.815 ± 0.936	8.856 ± 0.935	6.739 ± 0.857	
	Low	8.462 ± 1.000	8.119 ± 0.968	8.948 ± 0.960	6.655 ± 0.892	
NIE	High	9.907 ± 0.935	8.894 ± 0.927	10.024 ± 0.881	7.752 ± 0.737	
	Low	9.752 ± 1.242	9.282 ± 1.210	10.318 ± 1.212	6.862 ± 0.867	

Notes: * *p* < 0.05, ** *p* < 0.01. (a) Examining differences in types of conditions at the electrode points; (b) examining differences in risk levels at the electrode points.

**Table 3 brainsci-13-00486-t003:** Summary of event-related potential (ERP) results.

Conditions	NE		PIE		NIE		Pairwise Comparison Results
Risk Level (High/Low)	High	Low	High	Low	High	Low	
P150	−0.328 ± 0.549	1.001 ± 0.587	−0.448 ± 0.425	0.679 ± 0.482	−0.640 ± 0.478	0.482 ± 0.516	(b) High-risk < Low-risk (*p* = 0.000 ***)
N170	−1.774 ± 0.733	−1.521 ± 0.781	−1.806 ± 0.804	−1.153 ± 0.727	−1.537 ± 0.816	−0.845 ± 0.856	(b) High-risk > Low-risk (*p* = 0.021 *)
P2	6.549 ± 0.838	6.155 ± 0.835	6.545 ± 0.828	6.572 ± 0.867	7.339 ± 0.810	7.374 ± 0.987	-
N2	−4.667 ± 0.672	−4.673 ± 0.650	−4.683 ± 0.499	−4.930 ± 0.435	−5.354 ± 0.452	−5.696 ± 0.471	(a) NIE > PIE (*p* = 0.042 *)
LPP	−0.636 ± 0.725	−0.325 ± 0.725	2.755 ± 6.908	18.401 ± 23.601	−5.618 ± 5.333	−17.602 ± 7.234	(a) NIE < NE (*p* = 0.039 *)

Notes: * *p* < 0.05, *** *p* < 0.001. (a) Examining differences in types of conditions at the electrode points; (b) examining differences in risk levels at the electrode points.

## Data Availability

Not applicable.

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
