# Peer review of "The Influencing Mechanism of Incidental Emotions on Risk Perception: Evidence from Event-Related Potential"

_brainsci, 2023, doi:10.3390/brainsci13030486_

Round 1

Reviewer 1 Report

Comments and Suggestions for Authors

The manuscript “The influencing mechanism of incidental emotions on risk perception: Evidence from event-related potential” by Zhang et colleagues provides an interesting systematic approach to the characterization of incidental emotions on perception and decision making.

The manuscript is interesting, and it can be a positive contribution for the readers of “Brain Sciences”. However, the authors must completely revise the manuscript to improve the transparency and the quality of theoretical and methodological backgrounds.

1.       The manuscript is organized in a dysfunctional fashion. The authors should rewrite/rearrange it because in the current state it is not appropriately reviewable.

a.       The section "Introduction" describes the topic of the study and reports some literature. This section is strangely followed by a second introductory section labelled "Literature Review". This makes the manuscript confusing and less readable. Please elaborate a unique section ("Introduction") in which an accurate literature review  should also be included. The text in the actual section 2 should be shortened and condensed. Redundancies should be avoided. Please be sure to include in the Introduction also the specific aims of the study and the experimental hypothesis.

b.       Part of the text in the Methods section does not belong to the Methods, but to the Results section. For example, L276: “Results showed that the emotions evoked in experiments were consistent with those 276 measured in the pre-test” are Results.
Another example: in L306-308 “As shown in Figure.2(A), P150/N170, P2, N2, and LPP components were elicited in all experiments. Based on Figure.2(A) and Figure.2(B), it can be observed that the P150/N170 complex wave” there the authors are describing EEG results.

c.       Paragraph 4.3. “Construction of emotional information processing model”. These are Methods? Why is a model described between Results & Discussion? This does not make sense.

2.       The authors did not appropriately consider empirical evidence on cognitive and emotional interference (e.g., in Lines 55-77; Lines ). For example, regional deactivation/activation patterns and cross-network functional connectivity is crucial to maintain optimal performance, and the balance of activation/deactivation/connectivity seems to be specific for cognitive versus emotional interference and task-specific, mainly involving dynamics between the default mode network and other brain networks (Leech et al., 2014, PLoS One, DOI: 10.1371/journal.pone.0098500; Di Plinio et al., 2018, Hum Brain Mapp, DOI: 10.1002/hbm.24197; Codispoti et al., 2016, CABN, DOI: 10.3758/s13415-016-0453-0; Nee et al., 2013, Cereb Cortex,DOI: 10.1093/cercor/bhs007). Please consider these important findings when interpreting & discussing the influence of incidental emotions in your study.

3.       The grammar should be improved. By skimming the text, there many errors and bad phrasing which make the text less intelligible. Please note that the following are just some examples.

a.       L37. “The mechanism of the role of” should be "The role of".

b.       L51. “Then” should be “Thus”.

c.       L58. “non-invasiveness”.

d.       L64. “tasks”.

e.       L224-225 (and 234-235). Did the 20 participants rate the clips/pictures or not? Please be specific.

f.        L239-240. What does the numbers in parenthesis mean here?

g.       L246. Two groups of what?

h.       L261-265. The text here is not clear. Please explain better and use plain English.

i.         L271-277. It seems that the authors are reporting instructions to participants here.

4.       In the Introduction, Lines 51-54 are confusing. What is “process”? And, most importantly, the authors state they want to “try to reduce the incidence of safety accidents”. Are they sure that they are doing that? How?

5.       How was valence studied? Usually, low valence=Negative images and high valence=positive images, but results in line 230-231 are puzzling.

6.       From Lines 273-274, it seems that participants had to perform an action versus not-perform an action to distinguish two responses. This choice is inconvenient since it mingles task-effect with action inhibition and action initiation. It is more appropriate to have participants to choose between two actions (e.g. press index versus middle finger). Was at least this aspect counterbalanced across subjects?

Author Response

Response to Reviewer 1 Comments

We would like to thank the editor and the reviewers for their comments and their kind suggestions for our manuscript entitled “The influencing mechanism of incidental emotions on risk perception: Evidence from event-related potential” (ID: brainsci-2244683). The manuscript has been revised according to their comments and suggestions. We provide this cover letter to explain, point by point, the details of our revisions in the manuscript and our responses to the reviewer 1’s comments as follows. To make the changes easily viewable for the editor and the reviewers, in the revised paper, we marked the revision with red color. Besides, we have carefully checked through the whole manuscript and corrected some grammatical and referencing mistakes. The specific response is attached to the uploaded Word file.

Reviewer 2 Report

Comments and Suggestions for Authors

In this manuscript, the authors investigated the influence of "incidental emotions" on risk perception using an "oddball paradigm", the novelty of the study, as the authors argue, is the use of EEG data. The influence of mood on decision-making is a timely topic and the current manuscript may make an important contribution to the field. However, the reviewer finds the manuscript poorly written in many places. The authors should extensively revise the manuscript to facilitate understanding.

Major concerns

1), "incidental emotions" is not defined in the introduction, more importantly, the authors used videos to induce emotions, can these emotions be considered "incidental emotions"?

2), the authors mentioned they used an "oddball paradigm", this paradigm is not well explained in the methods section. what is it, how does it evaluate risk perception? what is the behavioral index of risk perception used in this study, these should be explicitly described in the methods section.

3), the abstract is generally hard to follow, the authors should make differences between results and discussion/interpretation, currently it is unclear which is result which is interpretation. not to say that some of the interpretations is problematic: shorter reaction time indicates faster risk perception? what is "a negative bias"? furthermore, oddball paradigm evaluates risk or risk perception or something else?

4), introduction: the authors keep saying that emotion affects risk perception, but without specifying what kind of influence it is. e.g., lines 42-43, lines 46-49.

line 45, by "the current task", do the authors mean driving? more explanation will facilitate understanding.

lines 55-57: the authors argue that behavioral experiments are limited by subjectivity, which is inappropriate. behavioral data are considered objective as well as EEG data.

lines 63-64: cognitive process does not equal to neuroimaging evidence, can the authors confirm the logic of this sentence?

lines 69-70: what kind of correlation it is here?

lines 71-73: what is "high-risk pictures" what is "emotional negativity bias"?

section 2.3, can the authors provide a figure explaining the components of EEG indices and what they indicate? this will greatly facilitate understanding.

5), methods:

sample size: why do the authors think 20 subjects is sufficient for the current analysis? how were the subjects recruited?

in section 3.1, the authors mentioned 20 subjects, however, in section 3.2.1, they mentioned 25, and in section 3.2.2 they mentioned 172, what is the big picture of this study? the authors should revise this part to avoid confusion and misunderstanding.

section 3.2.1, can the authors specify how was the valence and arousal ratings conducted, on what scale? this information will help understand the result reported in lines 230.

Fig 1: the image is the main message and should be enlarged, right now it is hard to see clearly. More importantly, can the authors enrich Fig 1 to illustrate the design of this study? they had three emotion states, and in each state, they had the trial of picture rating, fig 1 can be improved to show all these information to facilitate understanding of this study.

For EEG data, was there any noise detection and data cleaning procedure?

6), results:

Tables 1-3, before *, **, or ***, can the authors report the raw p value?

It seems to the reviewer that section 4.3 belongs to discussion, not results.

Author Response

Response to Reviewer 2 Comments

We would like to thank the editor and the reviewers for their comments and their kind suggestions for our manuscript entitled “The influencing mechanism of incidental emotions on risk perception: Evidence from event-related potential” (ID: brainsci-2244683). The manuscript has been revised according to their comments and suggestions. We provide this cover letter to explain, point by point, the details of our revisions in the manuscript and our responses to the reviewer 2’s comments as follows. To make the changes easily viewable for the editor and the reviewers, in the revised paper, we marked the revision with red color. Besides, we have carefully checked through the whole manuscript and corrected some grammatical and referencing mistakes. The specific response is attached to the uploaded Word file.

Round 2

Reviewer 2 Report

Comments and Suggestions for Authors

Thank the authors for addressing my concerns.